

# Ammonia-oxidizing bacterial communities are affected by nitrogen fertilization and grass species in native C$_4$ grassland soils

Jialin Hu[1], Jonathan D. Richwine[2], Patrick D. Keyser[2], Lidong Li[3], Fei Yao[1], Sindhu Jagadamma[1] and Jennifer M. DeBruyn[1]

[1] Department of Biosystems Engineering and Soil Science, University of Tennessee, Knoxville, TN, United States of America
[2] Department of Forestry, Wildlife and Fisheries, University of Tennessee, Knoxville, TN, United States of America
[3] Agroecosystem Management Research Unit, USDA-Agricultural Research Service, Lincoln, NE, United States of America

## ABSTRACT

**Background**. Fertilizer addition can contribute to nitrogen (N) losses from soil by affecting microbial populations responsible for nitrification. However, the effects of N fertilization on ammonia oxidizing bacteria under C$_4$ perennial grasses in nutrient-poor grasslands are not well studied.

**Methods**. In this study, a field experiment was used to assess the effects of N fertilization rate (0, 67, and 202 kg N ha$^{-1}$) and grass species (switchgrass (*Panicum virgatum*) and big bluestem (*Andropogon gerardii*)) on ammonia-oxidizing bacterial (AOB) communities in C$_4$ grassland soils using quantitative PCR, quantitative reverse transcription-PCR, and high-throughput amplicon sequencing of *amoA* genes.

**Results**. *Nitrosospira* were dominant AOB in the C$_4$ grassland soil throughout the growing season. N fertilization rate had a stronger influence on AOB community composition than C$_4$ grass species. Elevated N fertilizer application increased the abundance, activity, and alpha-diversity of AOB communities as well as nitrification potential, nitrous oxide (N$_2$O) emission and soil acidity. The abundance and species richness of AOB were higher under switchgrass compared to big bluestem. Soil pH, nitrate, nitrification potential, and N$_2$O emission were significantly related to the variability in AOB community structures ($p < 0.05$).

## INTRODUCTION

Nitrifiers play an important role in the productivity and sustainability of agricultural ecosystems. Nitrification is the biological oxidation of ammonia (NH$_3$) to nitrate (NO$_3^-$), carried out in a two-step process by nitrifiers. Ammonia-oxidizers [ammonia-oxidizing bacteria (AOB), ammonia-oxidizing archaea (AOA), and comammox bacteria] control the first and rate limiting step of nitrification: oxidation of NH$_4^+$ to NO$_2^-$ (*Daims et al., 2015*; *Frijlink et al., 1992*; *Venter et al., 2004*). Nitrous oxide (N$_2$O), a byproduct of ammonia oxidation (*Bakken & Frostegård, 2020*; *Hallin et al., 2018*), is a greenhouse gas with a global

Corresponding author
Jennifer M. DeBruyn,
jdebruyn@utk.edu

warming potential 265–298 times that of $CO_2$ and a major cause of the stratospheric ozone layer destruction (*Qin et al., 2014*; *Ravishankara, John & Robert, 2009*). About 56%–70% of the atmospheric $N_2O$ is contributed by microbial processes (such as nitrification and denitrification) in terrestrial ecosystems, with approximately 40% of terrestrial $N_2O$ emissions derived from agricultural soils (*Conrad, 1996*; *Davidson, 2009*; *Syakila & Kroeze, 2011*). In many cropping systems nitrification is the main source of $N_2O$ emissions from soils (*Liu et al., 2016*). $N_2O$ emissions from AOB activity is higher than from comammox bacteria and AOA because these later two groups lack homologues of AOB NO reductase (*Prosser et al., 2020*). In agricultural soils with N fertilization, nitrification can also lead to nitrate ($NO_3^-$) leaching (*Beeckman, Motte & Beeckman, 2018*), potentially contributing to eutrophication of nearby water systems (*Cameron, Di & Moir, 2013*). Major perennial forage crops prefer to take up N in the form of $NO_3^-$ rather than $NH_4^+$ (*Xu et al., 2019*), thus nitrifiers play a role in forage productivity.

AOB convert $NH_3$ to $NO_2^-$ through two enzymes: ammonia monooxygenase (AMO) and hydroxylamine oxidoreductase (HAO) (*Araki et al., 2004*; *Prosser, Microbiology & Group, 1986*) whereas AOA lack HAO homologs and may oxidize ammonia *via* nitroxyl (HNO) (*Levy-Booth, Prescott & Grayston, 2014*; *Schleper & Nicol, 2010*; *Walker et al., 2010*). Molecular ecology studies of AOA and AOB populations often use the marker gene *amoA*, which encodes α-subunit of AMO containing the active site (*Kowalchuk et al., 2000*; *Levy-Booth, Prescott & Grayston, 2014*; *Purkhold et al., 2000*; *Rotthauwe, Witzel & Liesack, 1997*). Although AOA appear to be more numerically abundant than AOB in some soils (*Leininger et al., 2006*), the abundance and community composition of AOB are more responsive to N fertilization, and therefore are functionally dominant for $NH_3$ oxidation in fertilized agricultural soils (*Di et al., 2009*; *Jia & Conrad, 2009*; *Ouyang et al., 2016*; *Segal et al., 2017*; *Ying et al., 2017*). The functional dominance of AOBs suggests this group should be the focus when developing strategies to control nitrification in agricultural soils. Many studies have demonstrated that the majority of known AOB in soil are *Nitrosospira* spp., with lesser contributions from *Nitrosococcus* and *Nitrosomonas* (*Lourenco et al., 2018*; *Pommerening-Roser & Koops, 2005*; *Stephen et al., 1996*).

Grasslands account for about 46.8% of all agricultural lands in the United States (USDA-NASS, 2012). Grasslands mobilize large pools of N with potential for N losses through $N_2O$ emissions and $NO_3^-$ leaching (*Di & Cameron, 2002*; *Duan et al., 2017*; *Francis et al., 1995*; *Hansen, Eriksen & Vinther, 2007*) with nitrification as a key control. Some grass species produce root exudates containing compounds that inhibit the enzymes AMO and HAO, known as biological nitrification inhibition (BNI) (*Subbarao et al., 2009*). BNI varies significantly among different grass species. A previous study has shown that $C_4$ grasses (*e.g.*, *Brachiaria* and *Sorghum*) exhibit greater BNI than $C_3$ grasses (*e.g.*, ryegrass and fescue) (*Subbarao et al., 2006*). Perennial native $C_4$ grasses, such as switchgrass (*Panicum virgatum* (SG)) and big bluestem (*Andropogon gerardii* (BB)), have been widely planted due to their drought-tolerance and high productivity under N-poor conditions (*Fike et al., 2006*; *Parrish & Fike, 2005*). A previous study reported that the optimum N rate for switchgrass to maximize dry matter yield (50 to 120 kg N ha$^{-1}$) was higher than that for big bluestem (45 to 90 kg N ha$^{-1}$) (*Brejda, 2000*), which may be due to the difference in

BNI between these two $C_4$ grass species and/or the greater N-use efficiency of big bluestem than switchgrass (*Friesen & Cattani, 2017*). Further, switchgrass contains a higher crude protein content compared to big bluestem (*Newell, 1968*) suggesting that switchgrass may provide more available N for AOB *via* plant residue decomposition. However, the AOB communities associated with these two $C_4$ grass species have not been well documented.

Studies have found that AOB abundances and activity increase with $NH_4^+$ addition, and correlate to nitrification rates (*Di et al., 2009*; *Smits et al., 2010*; *Wertz, Leigh & Grayston, 2012*), and that AOB community structure can be altered by N addition (*Avrahami, Liesack & Conrad, 2003*; *Rooney et al., 2010*; *Wertz, Leigh & Grayston, 2012*) and plant species (*Faulwetter et al., 2013*; *Rooney et al., 2010*) in other soils. However, there is a lack of understanding of the impact of N addition on AOB populations in native $C_4$ grasslands. To address this, our study aimed to: (i) determine the effects of N fertilization rate and grass species on AOB communities; and (ii) identify edaphic properties related to AOB abundance, activity, richness, evenness, and community composition, with a particular focus on the relationship of AOB with soil nitrification potential and $N_2O$ emissions. We hypothesized that: (1) N fertilization would promote AOB abundance, activity, richness and evenness, as well as nitrification potential and $N_2O$ emission; and a high N fertilization rate would alter AOB community structure; and (2) the abundance, activity, richness, and evenness of AOB would be higher under switchgrass than big bluestem because of differences in protein content and/or BNI between the grass species. This study was conducted in a native grassland plot experiment, and quantitative PCR (qPCR) of *amoA* genes from extracted DNA was used to estimate AOB abundances, quantitative reverse transcription PCR (qRT-PCR) of *amoA* genes from extracted RNA was used to estimate AOB activity, and high-throughput amplicon sequencing of AOB *amoA* genes was used to examine AOB alpha-diversity and community composition. AOB population data was then correlated to a suite of soil physicochemical properties, $N_2O$ emissions, and nitrification potentials.

## MATERIALS & METHODS

### Experimental design and sample collection

This study was conducted at a small-plot trial at the University of Tennessee East Tennessee Research and Education Center (ETREC) in Knoxville, TN (35.53°N, 83.06°W). Soils at this site are classified as sandy loam (fine-loamy, mixed, semiactive, thermic Typic Hapludults) with 59.9% sand, 23.5% silt, and 16.6% clay (*Bandopadhyay et al., 2020*). The experimental design was a randomized complete block with split-plot treatment arrangements. The main plots had two $C_4$ native grass species (switchgrass (SG) (*Panicum virgatum*) and big bluestem (BB) (*Andropogon gerardii*)) and the split-plots had different urea fertilization rates (0, 67, and 202 kg N ha$^{-1}$ (0N, 67N, and 202N, respectively)). Each combination of grass species and N rate had three replications. Grasses were planted in 2013, treatments were applied in 2014. Full experimental details, including the application of other fertilizers, lime, and herbicides were described in detail in our previous study (*Hu et al., 2021b*).

After five years of the experimental treatments, soil samples were collected as previously described (*Hu et al., 2021b*). Briefly, samples were collected at three grass growing seasons

in 2019: grass green up (G, late April, one week before the first N fertilization), initial grass harvest (H1, late June, within one week after harvest and before the second N fertilization), and second grass harvest (H2, Mid-August, within one week before harvest). Six soil cores (2.5 cm diameter, 10 cm long) were collected from each sub-plot and composited. A 10 g sub-sample was immediately stored at $-80\,°C$ for DNA and RNA extraction, and the remainder of the soil sample was used for soil physicochemical properties and nitrification potential measurements.

## Soil physicochemical analysis and nitrification potential

Soil pH, soil water content (SWC), $NH_4^+$-N, $NO_3^-$-N, and nitrification potential were analyzed using freshly collected soil. Total organic carbon (TOC), total nitrogen (TN), dissolved organic carbon (DOC), and dissolved organic nitrogen (DON) were analyzed using air-dried soils using established methods described in our previous study (*Hu et al., 2021b*). Soil nitrification potential was measured using a chlorate block method (*Belser & Mays, 1980*). Briefly, soil samples (2 g) were mixed with 14 mL nitrification potential medium (4 mL 0.2M $K_2HPO_4$, 0.5 mL 0.2M $KH_2PO_4$, 2.5 mL 0.2M $(NH_4)_2SO_4$ and 10 mL 1M $NaClO_3$ in 983 mL deionized water) in 50 ml tubes and shaken for 4-6 h (175 rpm, 25 °C). The nitrification potential medium is a solution containing adequate $NH_4^+$ and $PO_4^{3-}$ to maximize nitrification rates along with chlorate to block nitrite oxidation to nitrate. one mL slurry subsamples were removed at 3 h intervals and centrifuged to remove soil particles. $NO_2^-$ concentrations were determined by adding 50 $\mu$L of $NO_2^-$ color reagent (4 g of sulfanilamide and 0.1 g N-(1-naphthyl)-ethylenediamine hydrochloride and 17 mL 85% phosphoric acid in 87.6 mL deionized water) to 200 $\mu$L of centrifuged samples in a 96-well plate. After 15 min, the absorbance was measured in a microplate reader spectrophotometrically at $\lambda = 543$ nm. We used absorbance values for the $NO_2^-$ standards to compute the $NO_2^-$ concentration for each sample, subtracting the non-soil controls. Nitrification potential was calculated as $\mu$g $NO_2^-$-N $g^{-1}$ dry weight soil $hour^{-1}$ ($gdw^{-1}$ $h^{-1}$).

## Soil $N_2O$ emissions

Soil $N_2O$ emissions were measured using the static chamber method (*Collier et al., 2014*) on the same days that soil samples were taken. A circular stainless-steel base (15 cm in diameter, 16 cm in height) was inserted into the soil to a depth of approximately eight cm at the center of each sub-plot. The chamber was closed with a lid and gases were sampled through a valve in the lid with plastic syringes (30 mL) at four-time intervals (0, 20, 40, and 60 min). The samples were transferred to pre-evacuated glass vials (12 mL). Soil temperature was measured by inserting a digital thermometer (Thermo Scientific, USA) one inch into the soil adjacent to the chamber.

Gas samples were analyzed in a gas chromatograph (Shimadzu GC-2014, Japan) with an electron capture detector for $N_2O$ determination. $N_2O$ flux was calculated by linear interpolation of the four sampling times. The $N_2O$-N emission flux was calculated as follows:

$$F = \rho \times h \times dc/dt \times 273/(273 + T) \times k$$

where $F$ is emission flux ($\mu$g N m$^{-2}$ h$^{-1}$); $\rho$ is the N$_2$O-N density in standard state (1.25 kg m$^{-3}$); $h$ is the net height of static chamber (0.08 m); d$c$/d$t$ is the rate of gas concentration change ($\mu$L L$^{-1}$ min$^{-1}$); $T$ is the average soil temperature in the static chamber during gas sampling (°C); $k$ is the time conversion factor (60 min h$^{-1}$).

## Soil nucleic acid extraction and cDNA synthesis

Nucleic acid extraction and cDNA synthesis methods were described in our previous study (*Hu et al., 2021b*). Briefly, soil genomic DNA and RNA were extracted using the DNeasy PowerSoil Kit and RNeasy PowerSoil Total RNA Kit (Qiagen, Hilden, Germany), respectively, according to the manufacturer's protocols. DNA and RNA concentrations were quantified using the NanoDrop One spectrophotometry (NanoDrop Technologies, Wilmington, DE). RNA was checked for DNA contamination by attempting to PCR amplify *amoA* genes; samples that produced a negative electrophoresis result (*i.e.,* no DNA contamination) were used for reverse transcription. cDNA was produced with RNA as the template using SuperScript IV Reverse Transcriptase and random hexamer primers (Invitrogen, Paisley, UK) according to the manufacturer's protocol. RNA was stored at $-80$ °C; DNA and cDNA samples were stored at $-20$ °C.

## Quantitative analysis of *amoA* genes and gene transcripts

Quantitative PCR (qPCR) of AOB *amoA* genes and transcripts were performed on a CFX96 Optical Real-Time Detection System (Bio-Rad, Laboratories Inc., Hercules, CA, USA) using the primer pair *amoA*-1F (5′- GGGGTTTCTACTGGTGGT -3′) and *amoA*-2R (5′- CCCCTCKGSAAAGCCTTCTTC -3′) (*Rotthauwe, Witzel & Liesack, 1997*) as we have previously described (*Hu et al., 2021a*).

## *amoA* gene amplicon sequencing

A two-step PCR was used for *amoA* amplicon sequencing library preparation. We used methods similar to our previous study on *nifH* genes (*Hu et al., 2021b*) for the *amoA* genes. First, *amoA* genes were amplified from diluted DNA extracts (10 ng $\mu$l$^{-1}$) with primers *amoA*-1F/*amoA*-2R containing Illumina-compatible adapters (*Klindworth et al., 2013*). The amplifications were performed in a 25-$\mu$l mixture containing 12.5 $\mu$l KAPA HiFi HotStart ReadyMix (2×) (Kapa Biosystems), 1 $\mu$l of each primer (final concentration of 0.4 $\mu$M), 2.5 $\mu$l of template DNA, and 8 $\mu$l of PCR grade nuclease-free water. The PCR protocol was: 95 °C for 3 min, then 35 cycles of 98 °C for 20 s, 58 °C for 15 s and 72 °C for 15 s, followed by 72 °C for 5 min. PCR products were purified using SparQ PureMag Beads (Quantabio). For the second step, index PCR was performed in a 50-$\mu$L reaction containing 25 $\mu$l KAPA HiFi HotStart ReadyMix (2×) (Kapa Biosystems), 5 $\mu$l of each forward/ reverse NexteraxT index/primer, 5 $\mu$l of PCR product from the first step as template DNA, and 10 $\mu$l of PCR grade nuclease-free water. The PCR protocol was: 95 °C for 3 min, then 8 cycles of 95 °C for 30 s, 55 °C for 30 s and 72 °C for 30 s, followed by 72 °C for 5 min. PCR products were purified using SparQ PureMag Beads, quantified by NanoDrop, pooled, and run on an Agilent Bioanalyzer to ensure quality (*Hu et al., 2021b*). $2 \times$ 275-bp paired-ends sequencing was run on an Illumina Miseq platform (Illumina, CA, U.S.) in the University of Tennessee Genomics Core laboratory.

## Bioinformatics analysis

*amoA* sequences were processed using Mothur v.1.39.5 (*Schloss et al., 2009*). Forward and reverse reads were merged using make.contigs. Primer sequences were trimmed using trim.seqs. Reads were removed using screen.seqs if the overlap region of the merged reads was less than 10 bp, if the reads were shorter than 400 bp, or if the reads contained any ambiguous bases. Similar to the pipeline used in our previous study (*Hu et al., 2021b*), chimera.uchime was used to remove chimeras based on a manually created database containing 9682 AOB *amoA* gene reference sequences FunGene (http://fungene.cme.msu.edu/) (*Fish et al., 2013*). For all downstream analyses, all libraries were subsampled to a uniform sequencing depth (4,307 reads). Operational taxonomic units (OTUs) were classified based on 97% nucleotide sequence similarity (*Wang et al., 2014*; *Zheng et al., 2014*) using vsearch in QIIME2-2019.7 (*Bolyen et al., 2019*). Singletons (OTUs appearing only once across all samples) were removed, and representative sequences for each OTU were selected. Abundant OTU representative sequences (total reads ≥ 100) were blasted against the FunGene AOB *amoA*-like database using BLASTn. The raw sequencing data can be accessed from NCBI Sequence Read Archive (SRA) Database with BioProject accession number PRJNA733230.

## Statistical analysis

The statistical analyses used in this research were similar to our previous study (*Hu et al., 2021b*). Diversity analyses of AOB community were performed with QIIME2-2019.7. Chao1, observed OTUs, Shannon index, and Pielou's evenness were calculated to determine alpha-diversity (*Gourmelon et al., 2016*; *He et al., 2013*). A weighted-UniFrac method (*Lozupone et al., 2007*) was used for analyzing beta-diversity, similar to our previous study (*Hu et al., 2021b*). Permutational multivariate analysis of variance (PERMANOVA) was used to examine beta-diversity by treatment factors (*Hu et al., 2021b*). A neighbor-joining phylogenetic tree was constructed by QIIME using representative sequences of 28 abundant OTUs (OTUs with total reads ≥ 100) and was visualized using Interactive Tree of Life (iTOL, v5) (*Letunic & Bork, 2011*) as described in our previous study (*Hu et al., 2021b*). Principal coordinates analysis (PCoA) and redundancy analyses (RDA) based on weighted-UniFrac distances was performed in R 3.6.1 (The R Foundation for Statistical Computing) with packages vegan (2.5–5) and phyloseq.

A mixed model ANOVA within the GLIMMIX procedure in SAS 9.4 (SAS Inst., Cary, NC, USA) was performed to test treatment effects on soil properties, abundance and expression of *amoA* genes, and alpha-diversity of AOB communities, similar to our previous approach (*Hu et al., 2021b*). Gene and transcript abundances were log transformed to achieve normal distributions. Fixed effects included sampling season, N fertilization rate, and grass species, as well as their interactions. Random effects included the block and block by grass species. A least significant difference (LSD) method was used to compare the means of each two groups. Pearson correlation analyses were performed in IBM SPSS Statistics v26 (*Hu et al., 2021b*). IBM SPSS Amos 27.0 software was used to perform structural equation modeling (SEM) to characterize interactions among measured variables. We followed the procedures of developing and modifying a structural equation model in *Li et al. (2019)* and *Hu et al.*

*(2021b)*. Because community is a theoretical concept and not directly measurable, the "AOB community" component of the SEM model was a latent variable inferred from the measured relative abundances of the top 28 most abundant *amoA* OTU. These were OTUs with ≥ 100 reads, which comprised 97.9% of the total reads and were were classified into four species: *Nitrosospira* sp. 56-18, *Nitrosospira* sp. APG3, *Nitrosospira multiformis*, and *Nitrosospira* sp. Nsp14. For grass species, we defined switchgrass = 1, big bluestem = 2 due to the higher biomass of big bluestem than switchgrass. A well fit model should have a Chi-square/Df value in the range of 1.0 to 3.0 with probability level > 0.05, RMSEA value < 0.05, and fit indices (TLI, IFI, and CFI) both > 0.95 (*Blunch, 2012*; *Hooper, Coughlan & Mullen, 2008*; *Kline, 2015*).

# RESULTS

## Soil physicochemical characteristics, $N_2O$ emission, and nitrification potential

The soil physicochemical characteristics under different treatment combinations were described in detail in our previous study (*Hu et al., 2021b*). Briefly, SWC, DOC, and DON showed significant differences by sampling season ($p < 0.001$ for SWC and DOC; $p = 0.006$ for DON) (Table 1; Table S1). SWC was highest ($0.30 \pm 0.01$ g $H_2O$ $g^{-1}$ dry weight soil ($gdw^{-1}$)) at initial grass harvest (H1) and lowest ($0.15 \pm 0.01$ g $H_2O$ $gdw^{-1}$) at second grass harvest (H2). DOC and DON were highest ($223.00 \pm 5.47$ µg $gdw^{-1}$ and $22.44 \pm 1.98$ µg $gdw^{-1}$, respectively) at H2 and lowest ($163.85 \pm 4.07$ µg $gdw^{-1}$ and $16.16 \pm 0.65$ µg $gdw^{-1}$, respectively) at H1 (Table S1) (*Hu et al., 2021b*).

Nitrogen fertilization rate and $C_4$ grass species affected several soil properties (Table 1). With increased N fertilization rate, $NO_3^--N$ increased from $0.39 \pm 0.16$ to $2.77 \pm 0.39$ µg $gdw^{-1}$, soil pH decreased from $6.36 \pm 0.05$ to $6.12 \pm 0.05$, and DOC decreased from $210.18 \pm 8.60$ to $186.07 \pm 5.23$ µg $gdw^{-1}$. Soil pH was higher under switchgrass ($6.32 \pm 0.05$) compared to big bluestem ($6.19 \pm 0.03$). Fertilized plots had higher SWC and lower C:N for switchgrasss than 0N plots. In contrast, fertilization of big bluestem resulted in lower SWC and no change in C:N (Table S1).

$N_2O$ emission rate was affected by the interaction of sampling season, N fertilization rate, and grass species (Table 1; Table S1): at H1, $N_2O-N$ increased with N-rate, from $10.01 \pm 2.00$ to $23.77 \pm 2.08$ g $N_2O-N$ $ha^{-1}d^{-1}$ in switchgrass plots and from $8.74 \pm 3.92$ to $74.98 \pm 29.33$ g $N_2O-N$ $ha^{-1}d^{-1}$ in big bluestem plots; but at G and H2, no effect of N fertilization rate and grass species on $N_2O-N$ was observed.

There was a significant interaction effect between sampling season and N fertilization rate on nitrification potential (Table 1; Table S1): at G, nitrification potential was not significantly different under different N-rates, but at H1 and H2, nitrification potential was significantly higher under 202N. Nitrification potential was significantly higher in switchgrass plots ($0.14 \pm 0.02$ µg $NO_2-N$ $gdw^{-1}h^{-1}$) than in big bluestem plots ($0.10 \pm 0.02$ µg $NO_2-N$ $gdw^{-1}h^{-1}$) (Fig. 1; Table S1).

**Table 1 Results of mixed model ANOVA (based on GLIMMIX procedure in SAS) testing effects of sampling season, nitrogen fertilization and grass species on soil properties and *amoA* gene and transcript abundances.** *F*-values are reported.

| Effect | Season (S) | Nitrogen (N) | Grass (G) | N × G | S × N | S × G | S × N × G |
|---|---|---|---|---|---|---|---|
| pH | 3.26 | 11.70[***] | 11.53[**] | 0.06 | 0.59 | 1.99 | 0.13 |
| SWC[a] | 388.39[***] | 0.31 | 0.01 | 3.75[*] | 0.89 | 0.35 | 0.57 |
| $NH_4^+$-N | 23.06[***] | 0.01 | 0.04 | 1.09 | 3.40[*] | 3.05 | 1.35 |
| $NO_3^-$-N | 9.00[***] | 31.14[***] | 1.95 | 0.17 | 1.41 | 5.42[**] | 1.61 |
| DOC | 57.99[***] | 7.71[**] | 0.2 | 2.62 | 1.48 | 0.98 | 0.21 |
| DON | 6.10[**] | 1.2 | 1.91 | 1.06 | 1.64 | 0.83 | 1.13 |
| TOC | 0.43 | 0.88 | 0.11 | 1.13 | 1.58 | 1.22 | 0.84 |
| TN | 1.11 | 2.44 | 0.82 | 0.08 | 1.79 | 1.52 | 0.53 |
| C: N ratio | 0.98 | 2.15 | 2.01 | 4.35[*] | 0.24 | 0.24 | 0.26 |
| $N_2O$-N | 12.46[***] | 5.94[**] | 2.25 | 2.38 | 5.36[**] | 1.78 | 3.00[*] |
| Nitrification potential | 31.13[***] | 21.85[***] | 10.44[**] | 0.28 | 2.72[*] | 0.13 | 0.3 |
| *amoA* genes | 91.22[***] | 15.72[***] | 8.33[**] | 1.13 | 0.34 | 1.56 | 0.25 |
| *amoA* transcript | 17.20[***] | 8.30[**] | 0.67 | 1.44 | 3.33[*] | 0.92 | 0.48 |

**Notes.**

Significance level: * $0.01 < p\text{-value} \leq 0.05$; ** $0.001 < p\text{-value} \leq 0.01$; *** $p\text{-value} \leq 0.001$.

[a]SWC, soil water content; DOC, dissolved organic carbon; DON, dissolved organic nitrogen, TOC, total organic carbon; TN, total nitrogen.

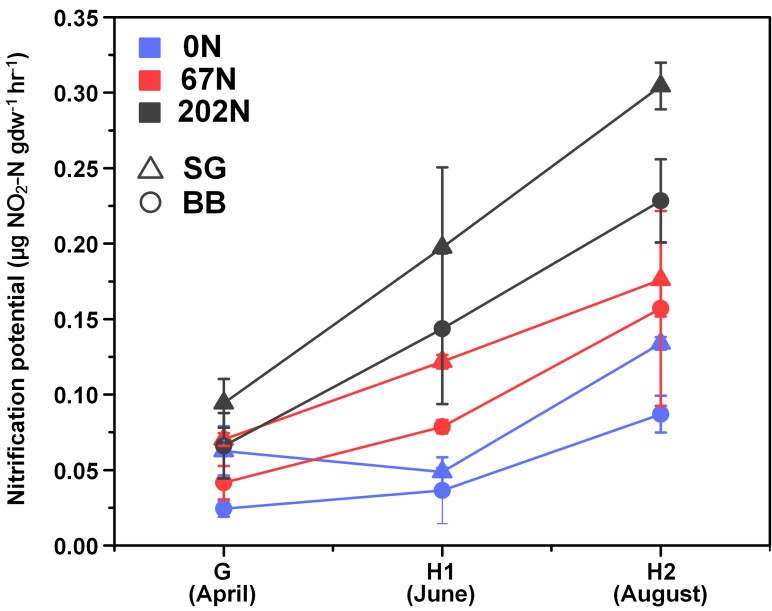

**Figure 1 Seasonal dynamics of nitrification potential in relation to nitrogen fertilization rate and grass species.** G, grass green up; H1, initial grass harvest; H2, second grass harvest; 0N, no N fertilization; 67N, 67 kg N ha$^{-1}$ fertilization; 202N, 202 kg N ha$^{-1}$ fertilization; SG, switchgrass; BB, big bluestem.

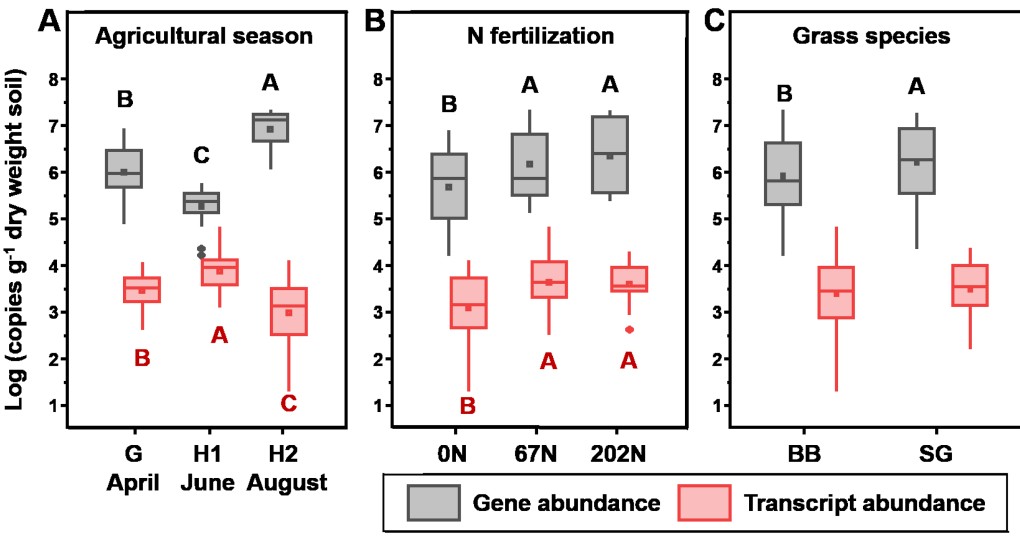

**Figure 2** **Absolute abundance of *amoA* gene (gray) and transcript (red) in relation to (A) sampling season; (B) nitrogen fertilization rates; and (C) grass species.** Boxes represent the interquartile range (IQR). Lines within boxes represent median values. Whiskers represent the range between minimum and maximum values. Diamonds represent outliers outside of this range. Dots within boxes represent mean values. Different letters above (black, for gene abundance) or below (red, for transcript abundance) the boxes indicate significant differences between treatment levels within groups by comparing means using Least Significant Difference (LSD) tests ($\alpha = 0.05$). The letters were only shown for groups with significant differences. G, grass green up; H1, initial grass harvest; H2, second grass harvest; 0N, no N fertilization; 67N, 67 kg N ha$^{-1}$ fertilization; 202N, 202 kg N ha$^{-1}$ fertilization; SG, switchgrass; BB, big bluestem.

## Abundances of *amoA* genes and transcripts

*amoA* gene abundances significantly varied with N fertilization rate, and grass species (Table 1). In general, *amoA* gene abundances were lowest at H1 ($2.51 \times 10^5$ copies g$^{-1}$ dry weight soil (gdw$^{-1}$)) and highest at H2 ($1.14 \times 10^7$ copies gdw$^{-1}$) (Fig. 2A). N fertilization increased *amoA* gene abundance ($5.03 \times 10^6$ and $7.00 \times 10^6$ copies gdw$^{-1}$ for 67N and 202N, respectively) compared to no N addition ($1.70 \times 10^6$ copies gdw$^{-1}$) (Fig. 2B). In addition, *amoA* genes were more abundant in switchgrass plots ($5.34 \times 10^6$ copies gdw$^{-1}$) than in big bluestem plots ($3.81 \times 10^6$ copies gdw$^{-1}$) ($p = 0.0066$) (Fig. 2C).

*amoA* gene transcript abundances were significantly affected by the interaction effect of sampling season and N fertilization rate (Table 1): at G, *amoA* transcript abundance was not different between N-rates; at H1, *amoA* transcript abundance was highest under 67N; at H2, *amoA* transcript abundances increased with increasing fertilizer. Grass species did not affect *amoA* transcript abundance (Table 1 and Fig. 2C).

## Ammonia-oxidizing bacterial structure and community

Using a 97% sequence similarity cutoff, a total of 215,943 high-quality bacterial *amoA* gene sequences were clustered into 538 OTUs, with 28 to 134 OTUs per sample. N fertilization rate and grass species significantly affected the alpha-diversity of AOB community (Table S2). AOB species richness (observed OTUs, Chao1) was lower at H1, but alpha-diversity estimates were similar across sampling times (Fig. 3A). Observed

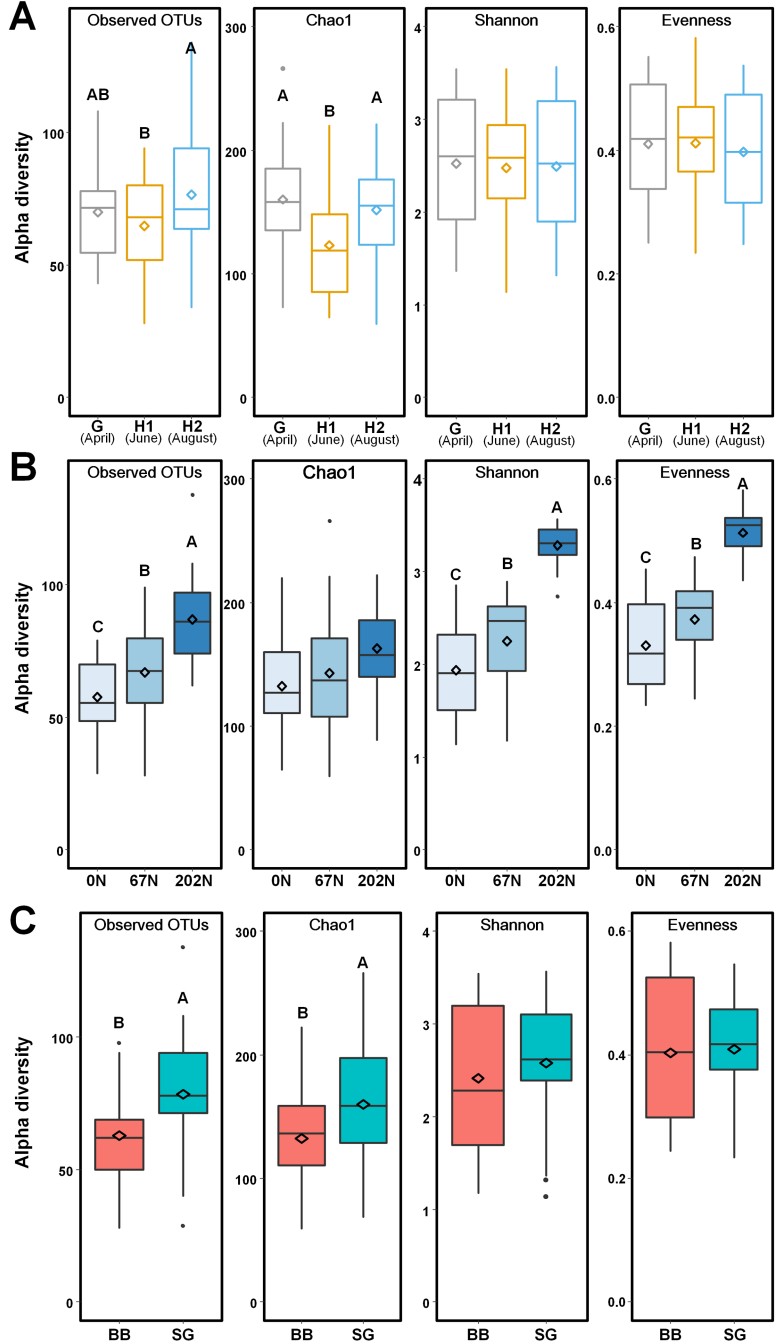

**Figure 3** **Alpha-diversity of ammonia-oxidizing bacterial communities in relation to (A) sampling seasons, (B) nitrogen fertilization rates, and (C) grass species.** Boxes represent the interquartile range (IQR). Lines within boxes represent median values. Whiskers represent the range between minimum and maximum values. Dots represent outliers outside of this range. Diamonds represent mean values. Different letters above boxes indicate significant differences between treatment levels within groups by comparing means using Least Significant Difference (LSD) tests ($\alpha = 0.05$). The letters were only shown in groups that have significant effects. G, grass green up; H1, initial grass harvest; H2, second grass harvest; 0N, no N fertilization; 67N, 67 kg N ha$^{-1}$ fertilization; 202N, 202 kg N ha$^{-1}$ fertilization; BB, big bluestem; SG, switchgrass.

**Table 2  Results of PERMANOVA examining the importance of the effects of season, N application rate, and grass species on the variation in ammonia-oxidizing bacterial community composition.**

| Factor | Df | F.Model | $R^2$ | *p*-value [a] |
|---|---|---|---|---|
| Season | 2 | 1.184 | 0.024 | 0.304 |
| Nitrogen | 2 | 24.706 | 0.504 | **0.001** |
| Grass | 1 | 4.101 | 0.042 | **0.021** |
| Season × Nitrogen | 4 | 0.425 | 0.017 | 0.931 |
| Season × Grass | 2 | 0.789 | 0.016 | 0.534 |
| Nitrogen × Grass | 2 | 1.640 | 0.033 | 0.164 |
| Season × Nitrogen × Grass | 4 | 0.661 | 0.027 | 0.741 |
| Residuals | 33 | | 0.336 | |

**Notes.**

[a] Bold values indicate *p*-value $\leq 0.05$.

OTUs, Shannon index, and Pielou's evenness increased with N fertilization rate (Fig. 3B). Higher richness was observed in switchgrass plots than in big bluestem plots (Fig. 3C).

AOB community structure was significantly affected by N fertilization rate and grass species (Table 2; Fig. 4). N fertilization rate had stronger effect (PERMANOVA $R^2 = 0.504$, $p = 0.001$) than grass species ($R^2 = 0.042$, $p = 0.021$) (Table 2). PCoA showed that AOB community composition in 202N was different from 0N and 67N treatments (PERMANOVA $p < 0.001$). Under 0N and 67N, the communities were much more variable under big bluestem compared to switchgrass (Fig. 4).

## Phylogenetic analysis of ammonia-oxidizing bacterial community

There were 28 abundant *amoA* OTUs (OTUs with $\geq 100$ reads) which comprised 97.9% of the total reads. A neighbor-joining phylogenetic tree showed that most of the 28 abundant *amoA* OTUs classified as *Nitrosospira*, with 8 OTUs (accounting for 11.8% of total reads) classified as *Nitrosospira multiformis* (Fig. 5). Many *amoA* OTUs were treatment-specific; for example, 8 OTUs (OTU4, OTU8, OTU9, OTU10, OTU12, OTU16, OTU19, and OTU22) classified as *Nitrosospira multiformis* and 2 OTUs (OTU11, OTU23) classified as *Nitrosospira* sp. 56-18 were more abundant in 202N compared to 67N or 0N. OTU19 was only detected in big bluestem plots with 202N, whereas OTU3 and OTU6, belonging to *Nitrosospira* sp. Nsp14, were more abundant under 0N and 67N in big bluestem plots. OTU2 (classified as *Nitrosospira* sp. Nsp14) had higher relative abundance under 0N and 67N in both switchgrass and big bluestem plots (Fig. 5).

## Relationships between soil properties and AOB populations

The abundances of *amoA* genes and transcripts were not correlated to each other (Table S3). The *amoA* gene abundance was negatively correlated to soil pH ($R = -0.303$, $p < 0.05$), SWC ($R = -0.792$, $p < 0.01$), $NH_4^+$-N ($R = -0.471$, $p < 0.01$) but positively correlated to DOC ($R = 0.477$, $p < 0.01$), DON ($R = 0.445$, $p < 0.01$), and nitrification potential ($R = 0.530$, $p < 0.01$). In contrast, the *amoA* gene transcript abundance was positively correlated to SWC ($R = 0.561$, $p < 0.01$), and available inorganic N ($NH_4^+$-N: $R = 0.448$, $p < 0.01$; $NO_3^-$-N: $R = 0.339$, $p < 0.05$), and negatively correlated to DOC ($R = -0.640$, $p < 0.01$) (Table S3). Alpha-diversity indices (except Chao1 index) were negatively

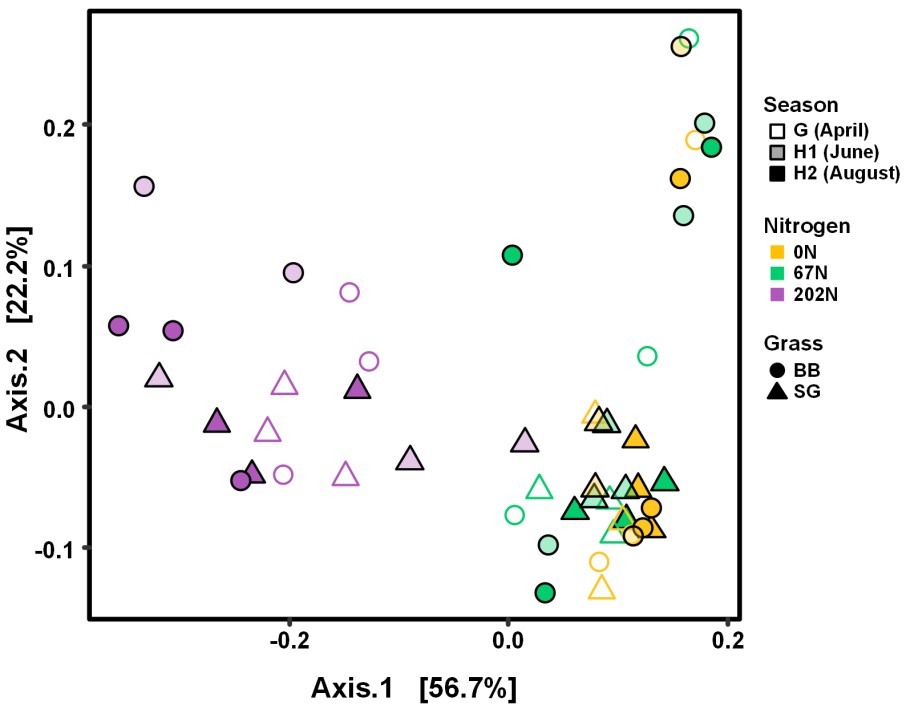

**Figure 4** PCoA of weighted-UniFrac distances between ammonia-oxidizing bacterial communities based on sampling season, N fertilization rate, and grass species. G, grass green up; H1, initial grass harvest; H2, second grass harvest; 0N, no N fertilization; 67N, 67 kg N ha$^{-1}$ fertilization; 202N, 202 kg N ha$^{-1}$ fertilization; SG, switchgrass; BB, big bluestem.

correlated to soil pH but positively correlated to $NO_3^-$-N and nitrification potential (Table S3). The Chao1 index, observed OTUs, and Shannon index were positively correlated to *amoA* gene abundance but not transcript abundance (Table S3).

Structural equation modeling (SEM) was conducted to identify the impacts of soil physicochemical parameters on AOB populations (Fig. 6; Table S4). SWC had a direct negative effect on *amoA* gene abundance (standardized path coefficient = −0.711, $p < 0.001$) and a direct positive effect on *amoA* transcript abundance (0.433, $p = 0.010$). *amoA* transcript abundance was directly and positively affected by $NH_4^+$-N concentration (0.331, $p = 0.002$) but directly and negatively affected by DOC (−0.493, $p < 0.001$). In addition, *amoA* gene abundance had a direct positive effect on its transcript abundance (0.541, $p < 0.001$). The Chao1 index was directly and negatively affected by $NO_3^-$-N (−0.366, $p < 0.001$), whereas the Pielou's evenness of AOB community was directly and positively affected by $NH_4^+$-N (0.268, $p < 0.001$). $N_2O$ emission rates were directly and positively affected by SWC (0.723, $p < 0.001$) and $NH_4^+$-N (0.167, $p = 0.022$). The total effects of N fertilization rate on *amoA* gene and transcript abundances, alpha-diversity of AOB community, nitrification potential and $N_2O$ emission rate were positive (Table S5). For the most abundant species within the AOB community, the total effects of N fertilization rate were positive for *Nitrosospira* sp. 56-18, *Nitrosospira multiformis*, and *Nitrosospira* sp. APG3 but negative for *Nitrosospira* sp. Nsp14. Compared to big bluestem, switchgrass

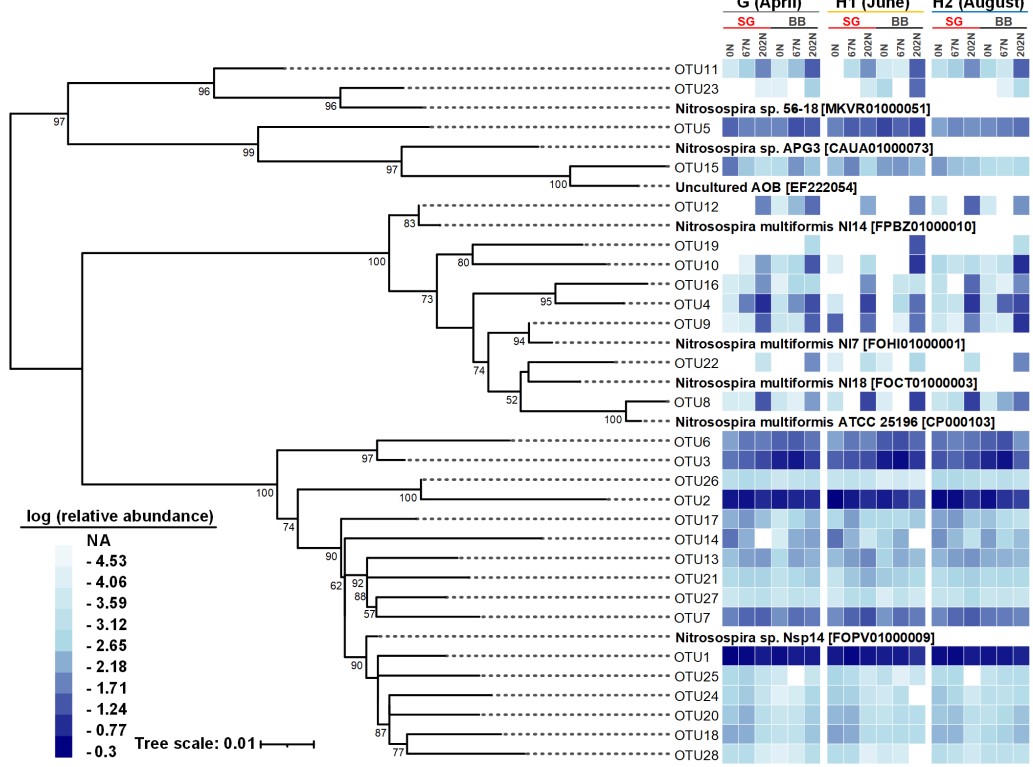

**Figure 5** **Neighbor-joining phylogenetic tree of the top 28 abundant *amoA* OTUs (OTUs with ≥ 100 reads).** Bootstrap values greater than 50% of 1,000 resamplings are shown near nodes. The scale bar indicates 0.01 nucleotide substitutions per nucleotide position. The blue-and-white heatmap shows the logarithmic value of OTU relative abundances in percentages in different treatment combinations of N fertilization rate and grass species in different sampling seasons. The NA in the legend means no reads were detected. GenBank accession numbers are shown for sequences from other studies. G, grass green up; H1, initial grass harvest; H2, second grass harvest; 0N, no N fertilization; 67N, 67 kg N ha$^{-1}$ fertilization; 202N, 202 kg N ha$^{-1}$ fertilization; SG, switchgrass; BB, big bluestem.

had higher *amoA* gene and transcript abundances, alpha-diversity of AOB community, nitrification potential, and soil $NO_3^-$-N content but grass species had no effect on AOB community composition (composed of the four *Nitrosospira* species mentioned above).

A redundancy analysis (RDA) showed that soil physicochemical properties, *amoA* gene and transcript abundances, sampling season, N fertilization rate, grass species, and alpha-diversity indices measured in this study explained a total of 62.5% of the variability of AOB community structures, with the first two axes explaining 50.2% of this variability (Fig. 7). Soil pH, $NO_3^-$-N, $N_2O$ emission, nitrification potential, *amoA* transcript abundance, N fertilization rate, grass species, and alpha-diversity significantly correlated to the variability of AOB community structures ($p < 0.05$) (Fig. 7).

## DISCUSSION

We investigated the impacts of N fertilization and native grass species on the seasonal population size, activity, alpha-diversity, and community structure of AOB. AOB

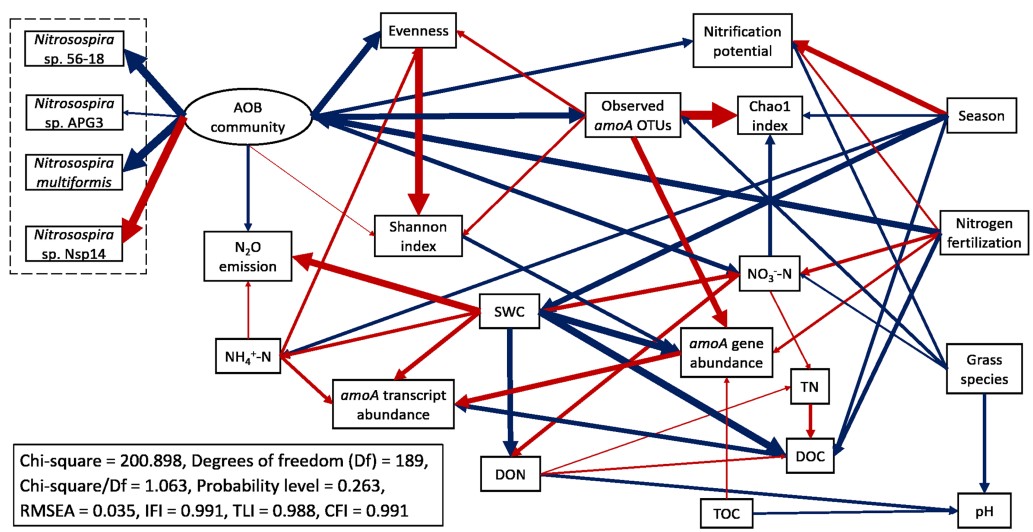

**Figure 6** **Structural equation modeling (SEM) for ammonia-oxidizing bacterial abundance, functional activity, and major community composition with key soil parameters.** Red arrows indicate positive relationships. Blue arrows indicate negative relationships. Single headed arrows represent causal relationship (*p*-value < 0.05). The direction of arrow indicates the direction of causation. The width of arrow indicates the extent of effects, *i.e.,* the standardized path coefficients are proportional to the thickness of arrows. See Table S4 for specific standardized path coefficients and Table S5 for standardized total effects. SWC, soil water content; DOC, dissolved organic C; DON, dissolved organic N; TOC, total organic C; TN, total N.

population size, activity, and species richness varied with sampling time, but AOB community structure was relatively constant. Nitrogen fertilization significantly promoted the abundance, activity, and alpha-diversity of AOB as well as nitrification potential, as it provided ammonium as initial substrate for nitrification process. Notably, the promotion of AOB activity by N addition was only observed during the grass growing season (June, August) and not at grass green up (April, before N fertilization), indicating that urea fertilization may only have short-term influence on AOB activity. Moreover, $NO_3^--N$, a product of nitrification, was closely and positively related to AOB activity and was significantly higher under high N-rate.

Our observations that N management can affect AOB community structure is consistent with previous studies (*Mendum & Hirsch, 2002*; *Rooney et al., 2010*; *Webster et al., 2005*). The influence of N fertilization on AOB communities can be partly explained by the lower soil pH at 202N (*Pommerening-Roser & Koops, 2005*; *Xi et al., 2017*): oxidation of ammonium produces $H^+$ ions that acidify the soil and may select for more acidic-tolerant AOBs. Indeed, in this study we observed a very different AOB community structure under high N-rate compared to moderate or no N fertilization. In particular, we found that OTUs classified as *Nitrosospira multiformis* were more abundant at the high urea fertilization rate. The genome of *N. multiformis* contains genes encoding urease, urea carboxylase, and a putative allophanate (carboxyurea) hydrolase for urea conversion to ammonium and bicarbonate (*Norton et al., 2008*). The ureolytic capacity of *N. multiformis* makes it well-adapted for soils undergoing urea fertilization and/or acidic pH

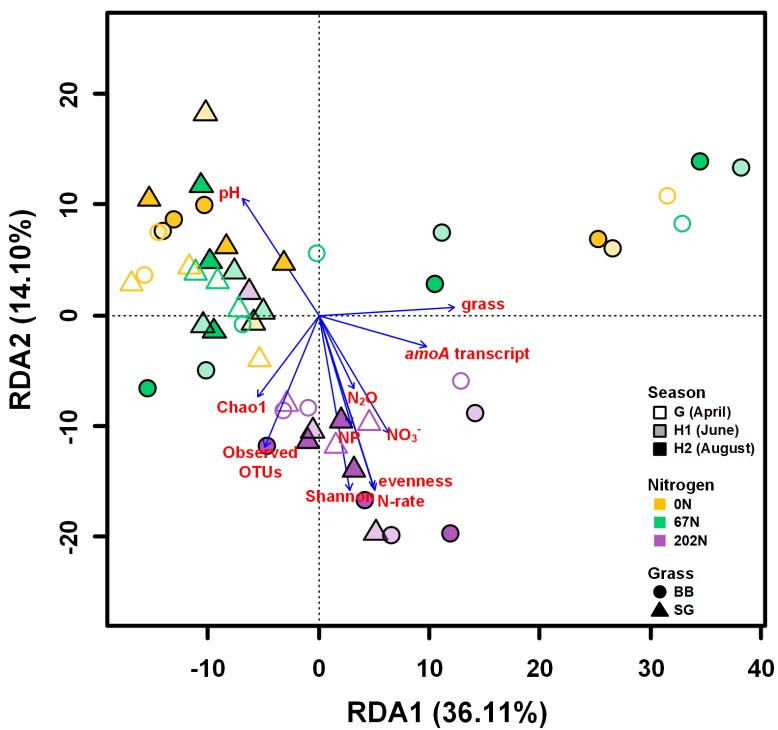

**Figure 7** **Redundancy analysis (RDA) of weighted-UniFrac distances between ammonia-oxidizing bacterial community structures (*amoA* OTU relative abundances).** Arrows represent loadings of environmental factors, *amoA* gene and transcript abundances, and community diversity metrics. G, grass green up; H1, initial grass harvest; H2, second grass harvest; 0N, no N fertilization; 67N, 67 kg N ha$^{-1}$ fertilization; 202N, 202 kg N ha$^{-1}$ fertilization; SG, switchgrass; BB, big bluestem; grass, grass species; N-rate, N fertilization rate; NP, nitrification potential.

(*Burton & Prosser, 2001*; *Pommerening-Roser & Koops, 2005*). Furthermore, the arginine decarboxylases in *N. multiformis* are known to function in acid-tolerance (*Norton et al., 2008*). It is unknown if the other *Nitrosospira* species we found in these soils contain urease and arginine decarboxylase.

AOB population size and species richness, as well as nitrification potential, were significantly higher under switchgrass compared to big bluestem. The AOB community structure was also significantly different between the two grasses. This suggests that plants may play important roles in shaping AOB communities and selecting distinct AOB populations. Root exudates of grasses contain active compounds, such as organic acids, which may inhibit nitrification (*Subbarao et al., 2009*), and differences in the types or concentrations of compounds from the two grass species may have influenced the soil microbial communities. Moreover, leached compounds from litter, such as phenolics, tannins or monoterpenes, may also have had different effects on: (1) the inhibition of ammonium monooxygenase (*Elroy & Sunil, 1973*; *White, 1986*); (2) ammonium immobilization by heterotrophs decreasing ammonium availability for AOB (*Bremner & McCarty, 1993*); or (3) the quality of soil organic carbon under different grass species (*Strauss & Lamberti, 2002*). Big bluestem has a higher cellulose content than switchgrass

(*Delaquis et al., 2014*), which suggest that we may see lower nitrification rates and/or AOB population size in switchgrass plots. However, our results showed that switchgrass had higher AOB abundance and nitrification potential as well as similar AOB activity compared to big bluestem. One possible explanation was that switchgrass tends to mature earlier than big bluestem (*Mitchell & Anderson, 2008*). When we sampled soils, the switchgrass had become stemmy and contained more cellulose as it matured (*Lemus, 2008*), which made its litter more difficult to decompose, and therefore reducing competition for ammonium from heterotrophic decomposers (*Strauss & Lamberti, 2002*). Lastly, the significant difference in AOB community structure but similar activity between switchgrass and big bluestem suggests that nitrification may be controlled by taxonomically different nitrifiers, and is indicative of functional redundancy within AOB communities (*Louca et al., 2016*; *Louca et al., 2018*; *Short et al., 2013*). It was notable that our SEM result predicted a lower abundance, activity, and alpha-diversity of AOB, as well as nitrification potential and soil nitrate content under big bluestem than under switchgrass, suggesting a lower potential for N losses through nitrate leaching and $N_2O$ emission in big bluestem systems.

AOB population dynamics were related to soil moisture in our study. The significant negative correlation of *amoA* gene abundance but positive correlation of *amoA* transcript abundance with SWC found in this study was also observed in a cotton field in the similar climatic zone (*Hu et al., 2021a*). One possible explanation was that increasing SWC decreases soil oxygen, resulting in decreased abundances of aerobic AOB. The increased activity we observed could be because AOB populations increased expression of *amoA* genes under oxygen stress. It has been found that the transcription of bacterial *amoA* genes can be promoted in soils with >70% water filled pore space (*Theodorakopoulos et al., 2017*). Another study also reported that low dissolved oxygen content can promote *amoA* gene transcription in *Nitrosomonas europaea* in culture (*Yu & Chandran, 2010*).

Dissolved organic C (DOC) is another key factor that may have influenced the variation of AOB abundance and activity. DOC was highest at second grass harvest, and negatively associated with SWC, indicating possible leaching loss or dilution of DOC under higher SWC conditions (*Poll et al., 2008*; *Zibilske & Makus, 2009*). Total precipitation during the initial harvest period (June, 161.29 mm) was higher than during grass green up and second harvest (April and August, 94.23 and 120.14 mm, respectively). Our study found that the population size of AOB increased with DOC concentration, which was also observed in some other agroecosystems (*Hu et al., 2021a*; *Sun et al., 2019*) and in groundwater (*Van der Wielen, Voost & vander Kooij, 2009*). One possible explanation might be that under DOC-rich conditions, mineralization of organic C by heterotrophs provided more $CO_2$ for the autotrophic AOB. However, there was a negative correlation between DOC and *amoA* transcript abundance in our study, which was also observed in estuarine ecosystems (*Happel et al., 2019*). Many studies have observed that nitrification rates can be inhibited by increased organic C concentration in terrestrial ecosystems due to (1) biological nitrification inhibition (*Elroy & Sunil, 1973*; *Janke, Wendling & Fujinuma, 2018*; *Subbarao et al., 2009*; *White, 1986*); (2) phenolics, tannins, and monoterpenes promoting heterotrophic immobilization of ammonium and decreasing substrate availability for nitrifiers (*Bremner & McCarty, 1993*; *Paavolainen, Kitunen & Smolander, 1998*); and (3)

the increased competition between heterotrophic and nitrifying bacteria for ammonium caused by high quality C (*i.e.,* labile carbon) (*Strauss & Lamberti, 2002*).

Similar to DOC, DON is also an important factor affecting AOB abundance and can be diluted or lost under high SWC conditions. There was a positive correlation between DON and AOB abundance in our study, which may be because DON, especially low molecular weight DON, provides initial substrate for nitrification (*Jones et al., 2004*). However, no correlation between DON and AOB activity was observed, indicating that the expression of *amoA* may be influenced by other climatic factors, such as temperature, soil texture, and other soil characteristics.

Nitrification potential was positively correlated to *amoA* gene abundances, confirming the reliability of using nitrification potential to infer the population size of nitrifiers (*Belser & Mays, 1980*; *Bernhard et al., 2010*). However, nitrification potential was not correlated to *amoA* transcript abundance, indicating that *in situ* AOB activity is likely influenced by climatic factors, nutrient content, and soil properties (*Lew, Glińska-Lewczuk & Lew, 2019*; *Nelson, Martiny & Martiny, 2016*). Although many studies have observed a positive correlation between the abundance of ammonia oxidizers and $N_2O$ emissions (*Breuillin-Sessoms et al., 2017*; *Pannu et al., 2019*), other studies, including ours, report no correlation of $N_2O$ emissions with AOB abundance and activity (*Lourenco et al., 2018*; *Pereira et al., 2015*). This is probably due to the fact that $N_2O$ emissions are regulated by both nitrification and denitrification processes through nitrifiers (AOB, AOA, and comammox bacteria) and denitrifiers (*Maag & Vinther, 1996*). We found that *amoA* transcript abundances significantly and positively associated to its gene abundance according to the SEM, indicating that AOB population size does affect its activity. Furthermore, the alpha-diversity of AOB communities was strongly and positively correlated to AOB population size and nitrification potential but had no correlation with AOB activity, suggesting that a more diverse AOB community may have more nitrification potential but may not influence *in situ* nitrification rate due to functional redundancy and environmental stress.

## CONCLUSIONS

We determined the dynamics of AOB population size, activity, alpha-diversity, and community structure under different N fertilization rates and $C_4$ grass species in a native perennial grassland. The genus *Nitrosospira* was the dominant AOB in this grassland. In accordance with our hypothesis, elevated N fertilizer application changed AOB community composition and increased abundance, activity, and alpha-diversity of AOB community as well as nitrification potential, $N_2O$ emissions, soil nitrate content, and soil acidity. This suggests high N addition in native $C_4$ grass systems may have negative consequences for N retention. The SEM predicted lower abundance, activity, and alpha-diversity of AOB, as well as nitrification potential and soil nitrate content under big bluestem compared to switchgrass, suggesting a lower potential for N losses. Together, our work revealed insights into important relationships among the abundance, activity, alpha-diversity, and community structure of AOB and their responses to N fertilization in perennial native $C_4$ grass systems.

## ACKNOWLEDGEMENTS

The authors are grateful to BJ Delozier, Cody Fust, Nicholas Tissot, Charles Summey, and Bobby Simpson and the staff of the East Tennessee Research and Education Center who manage and maintain the field trials, Sreejata Bandopadhyay for field sampling, Sutie Xu, Shikha Singh and Surendra Singh for data collection, and Mallari Starrett and Tori Beard for assistance in the laboratory.

### Funding

This work was supported by the United States Department of Agriculture (No. USDA-NIFA 2016-67020-25352). The funders had no role in study design, data collection and analysis, decision to publish, or preparation of the manuscript.

### Grant Disclosures

The following grant information was disclosed by the authors:
The United States Department of Agriculture: USDA-NIFA 2016-67020-25352.

### Competing Interests

The authors declare there are no competing interests.

### Author Contributions

- Jialin Hu performed the experiments, analyzed the data, prepared figures and/or tables, authored or reviewed drafts of the paper, and approved the final draft.
- Jonathan D. Richwine, Fei Yao and Sindhu Jagadamma performed the experiments, authored or reviewed drafts of the paper, and approved the final draft.
- Patrick D. Keyser conceived and designed the experiments, authored or reviewed drafts of the paper, and approved the final draft.
- Lidong Li analyzed the data, prepared figures and/or tables, authored or reviewed drafts of the paper, and approved the final draft.
- Jennifer M. DeBruyn conceived and designed the experiments, analyzed the data, authored or reviewed drafts of the paper, and approved the final draft.

### Data Availability

 The raw sequencing data is available at the NCBI Sequence Read Archive (SRA) Database: PRJNA733230.
 The raw qPCR data and soil properties data is available in the Supplementary Files.

### Supplemental Information

Supplemental information for this article can be found online at http://dx.doi.org/10.7717/peerj.12592#supplemental-information.

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
