# Peer review of "Ammonia-oxidizing bacterial communities are affected by nitrogen fertilization and grass species in native C4 grassland soils"

_PeerJ, doi:10.7717/peerj.12592_

## Round 0.1 · original submission · Minor Revisions

We have received three reports, all of them very positive regarding your research. Our reviewers do, however, find enough reasons for me to request improvement to your manuscript. Please address their comments, especially regarding the revision of the introduction and the apparent contradictions between lines 242-246, (“at G and H2, no effect of N fertilization rate and grass species on N2O-N was observed” ) and lines 434-436 ( “elevated N fertilizer application increased N2O emission” in lines ).

Reviewer 1 ·

Basic reporting

The manuscript by Hu et al. focused on the effects of N fertilization on AOB under C4 perennial grasses in nutrient-poor grasslands. The results showed that N fertilization rate had a stronger influence on AOB community composition than C4 grass species. Additionally, elevated N fertilizer application increased the abundance, activity, and alpha-diversity of AOB communities as well as nitrification potential, nitrous oxide (N2O) emission and soil acidity. Most importantly, soil pH, nitrate, nitrification potential, and N2O emission were significantly related to the variability of AOB communities. The topics is quite interesting, data are massive, and the manuscript is well organized. Thus, I recommended the manuscript to be published after a major revision.

Experimental design

None

Validity of the findings

1. Line 25, “ammonia oxidizers” not only include ammonia oxidizing bacteria, while the object of this study is ammonia oxidizing bacteria, so it should be replaced with “ammonia oxidizing bacteria”.
2. Line 37, Please be more specific about "Soil pH, nitrate, nitrification potential, and N2O emission were significantly related to the variability of AOB communities" Have you conducted the statistical analysis? If so, please add the p-value after "significantly". Additionally, the “variability of AOB communities” is too vague. Dose it mean the abundance or diversity?
3. Line 108, the author clarified that the experimental soil is sandy loam, however, the physical and chemical properties of soil should be added here.
4. Lines 169-171, please check and correct the initial references for AOB primers used in the experiment.
5. Lines 242-246, “at G and H2, no effect of N fertilization rate and grass species on N2O-N was observed” But, the author concluded that “elevated N fertilizer application increased N2O emission” in lines 434-436, which were contradictory. Additionally, according to the table S5, the relationship between “N fertilizer application” and “N2O emission” was weak. I suggest that the author clarify this problem.
6. Lines 432-433, “Nitrogen fertilization rate had a stronger influence on AOB community composition than C4 grass species” In this study, the amount of N fertilizer applied in experiment was artificially controlled, as well as the type of grass. How to compare the importance weights of the two kinds of independent variables? This conclusion is not universal and meaningless.
7. What do the letters "A, B, C" above and below the column in figure 2 refer to? Please clarify.
8. The author perform the structural equation modeling analysis and showed the results in figure 6. The “AOB community” and “grass species” are non-quantitative variables, how to conduct correlation and variance analysis? Please clarify.

Additional comments

None

Reviewer 2 ·

Basic reporting

The article was well written with sufficient literature cited to support the background and discussion as well as the methodology.

The structure of the article was well done in the expected style and format. Figures and tables were well designed and presented the key findings in a clear manner, with one exception. Figure 6 was not immediately clear to this reviewer when first examining it. My confusion came from the color scheme chosen--specifically the use of the color red to portray positive interactions. To me, red is normally associated with negative interactions, but after I realized what the color scheme was, the figure came into focus.

Sequence data was properly noted within the authors' declaration and within the paper. However, this reviewer was unable to find the files on NCBI and assumes that the authors have it embargoed until publication.

Experimental design

The experimental design was appropriate for the study.

No further comment.

Validity of the findings

No comment.

Reviewer 3 ·

Basic reporting

Hu et. al. examined the effects of nitrogen fertilization and grass species on ammonia-oxidizing bacteria in grassland soils, and revealed strongest impact of N fertilizer application rate on Nitrosospira-dominated AOB abundance and biodiversity. The manuscript is generally well written and provided some new perspectives on the potential influence of change in grass species on AOB composition and activity.
My biggest concern is about the introduction. The authors provide many misleading information and the references are often random and not the best. I show some examples below, but the authors should check this part thoroughly.
1) Line 42-44. Since the author mentioned AOA and AOB, the reference of Frijlink et al., 1992 cannot be enough for both (since first AOA was discovered after year 2000). A more recently review paper should be used to replace or complement this one.
2) Line 42-44. There is no mentioning comammox.
3) Line 49. Please read Prosser et. al. 2020 (Nitrous oxide production by ammonia oxidizers: Physiological diversity, niche differentiation and potential mitigation strategies) and other related papers for ecology of nitrous oxide emissions by ammonia oxidizers, especially by AOB, and provide some more specific information in the introduction, since AOB-driven N2O emission is an important result of this study.
4) Line 54-55. AOA do not contain HAO.
5) Line 62-64. I don’t think that is true. Maybe in some cases AOB are more abundant than AOA in agricultural topsoil, but that is not a leading trend. Also “AOB are more strongly correlated to nitrification rates than AOA” in agroecosystems mostly because they are “more responsive to N fertilization”, as the authors already described in previous sentence. Therefore, this statement should be deleted.
6) Line 65-67. Wrong information. There are tons of papers describing Nitrosospira as the primary AOB in soils, which is also the result of this study. Nitrosomonas is actually not commonly dominate AOB population in soil based on my experience, but more often observed in wastewater treatment plant. Please revise.
7) Line 84-85. Seems to repeat previous description (Line 58-60).
8) Line 95-96 How is hypothesis 2 generated. I suppose it has something to do with description in Line 73-82, but please provide additional explanation for a more direct link to this hypothesis.
9) Line 100. Community composition is also technically an index of diversity (beta-diversity). It might be best to just use richness and evenness to describe “diversity” here and in other part of this manuscript.

Experimental design

10) Line 133. To test soil nitrification rate or potential, it is most common to monitor nitrite plus nitrate concentration, because soil also contains nitrite oxidizers which convert nitrite to nitrate. Only measuring nitrite can lead to underestimation of nitrification potential. Please explain the method here.

Validity of the findings

no comment

Additional comments

11) Line 123. Add “was” before “used”.
12) Line 170. Please use the original papers as references for these primers, or don’t cite at all since you also cite your previous paper.
13) Line 198. Please use the original reference of QIIME. I don’t think the cited papers in 2014 used QIIME2. Please check all other programs and methodologies in the Materials & Methods for their proper references.
14) Line 257. Show switchgrass qPCR data in correspondence with big bluestem here.
15) Line 345-350. Have you confirmed that the other related Nitrosospira species in these soils do NOT contain urease and arginine decarboxylases?
16) Line 384-385. I think it is due to nitrifier denitrification of AOB under low oxygen content. But your soils should be “aerobic” enough.
17) Line 420-421. But the condition of your soils should not stimulate denitrification (see ref. Hink et al. 2017. Archaea produce lower yields of N2O than bacteria during aerobic ammonia oxidation in soil). Did you measure water-filled pore space in your soil to compare?

---

## Round 0.2 · accepted · Accept

Our reviewers are quite satisfied with your responses. I am glad to accept your paper for publication.

Reviewer 1 ·

Basic reporting

no comment

Experimental design

no comment

Validity of the findings

no comment

Additional comments

The revised manuscript has been well revised, clear presentation, sufficient references supporting. Published in its present version.

Reviewer 3 ·

Basic reporting

I am happy with the revisions and support the publication of this manuscript.

Experimental design

NA

Validity of the findings

NA

Additional comments

NA